# Mapping Land Use and Land Cover Classes in São Paulo State, Southeast of Brazil, Using Landsat-8 OLI Multispectral Data and the Derived Spectral Indices and Fraction Images

**Yosio E. Shimabukuro ***, **Egidio Arai, Gabriel M. da Silva**, **Tânia B. Hoffmann**, **Valdete Duarte,**
**Paulo R. Martini, Andeise Cerqueira Dutra**, **Guilherme Mataveli**, **Henrique L. G. Cassol**
**and Marcos Adami**

Instituto Nacional de Pesquisas Espaciais, Av. dos Astronautas, 1758, São José dos Campos 12227-010, Brazil; marcos.adami@inpe.br (M.A.)

\* Correspondence: yosio.shimabukuro@inpe.br

**Abstract:** This work aims to develop a new method to map Land Use and Land Cover (LULC) classes in the São Paulo State, Brazil, using Landsat-8 Operational Land Imager (OLI) data. The novelty of the proposed method consists of selecting the images based on the spectral and temporal characteristics of the LULC classes. First, we defined the six classes to be mapped in the year 2020 as forest, forest plantation, water bodies, urban areas, agriculture, and pasture. Second, we visually analyzed their variability spectral characteristics over the year. Then, we pre-processed these images to highlight each LULC class. For the classification, the Random Forest algorithm available on the Google Earth Engine (GEE) platform was utilized individually for each LULC class. Afterward, we integrated the classified maps to create the final LULC map. The results revealed that forest areas are primarily concentrated in the eastern region of São Paulo, predominantly on steeper slopes, accounting for 19% of the study area. On the other hand, pasture and agriculture dominated 73% of all São Paulo's landscape, reaching 39% and 34%, respectively. The overall accuracy of the classification achieved 89.10%, while producer and user accuracies were greater than 84.20% and 76.62%, respectively. To validate the results, we compared our findings with the MapBiomas Project classification, obtaining an overall accuracy of 85.47%. Therefore, our method demonstrates its potential to minimize classification errors and offers the advantage of facilitating post-classification editing for individual mapped classes.

**Keywords:** Land Use and Land Cover (LULC); forest; forest plantation; agriculture; pasture; urban; Linear Spectral Mixing Model (LSMM); spectral indices

## 1. Introduction

Important techniques for mapping and monitoring Land Use and Land Cover (LULC) changes are based on Geographic Information Systems (GIS) and remote sensing products obtained by orbital platforms. In recent decades, remote sensing has become an important tool to monitor the Earth's natural resources, since it provides data over large geographical extensions and in a temporal frequency that allows for the monitoring of the processes that occur in these areas [1].

Remote sensing images have greatly expanded the horizons of their applications in recent years due to the improvement of several factors such as number of satellites, sensor quality (radiometric and spatial resolutions), computational capacity, and higher availability of images freely distributed, allowing us to observe the Earth's surface in more detail and with higher frequency. The integration of these images with other spatial data sources (GIS) and the Geographic Database (GD) has significantly expanded the possibilities of spatial analysis of the Earth's surface data [2].

The mapping and monitoring of LULC are of great importance and there is an effort from the scientific community to provide and improve the accuracy of these products [3–5]. These studies aim to understand changes due to natural and anthropic factors [6,7] and improve efficiency in the elaboration of natural resource management plans [8,9], as well as the delimitation of priority areas for conservation [10]. LULC classification has a great impact on ecosystem management and has become an important base for much research into environmental hot topics worldwide [11–14].

In Brazil, the state of São Paulo is widely known as the largest national economic and industrial hub. In addition, it is a global leader in agribusiness, accounting for 13.5% of Brazilian crop production, while maintaining the largest number of Atlantic forest remnants [15,16]. The natural vegetation of the São Paulo State is mainly defined by the Atlantic Forest and Cerrado biomes, which have been massively removed since 1500 but have increased again in the last two centuries [16]. According to data from the Secretary of State for the Environment and Forest Inventory of São Paulo, it is estimated that São Paulo state has 24% remaining of the native vegetation of the Atlantic Forest and only 17% of the native vegetation of the Cerrado [16,17].

Long-term LULC data for the state of São Paulo and consequently those to quantify changes are important to understand current patterns of LULC and to improve planning and management of state resources and governance [3,18]. In addition, in order to understand the patterns and phytophysionomic diversity of the remaining vegetation, it is paramount to identify historical patterns and which factors, including relief shapes and topography, it might be related. The removal of native vegetation, mainly for agricultural purposes, has well-known alteration patterns, as well as the relief, water availability, and soil fertility, among other factors. Urban expansion also shows characteristic expansion patterns, with future expansion arcs in regions that are already urbanized.

In this way, understanding the historical characteristics of each LULC and linking them with the spectral and temporal responses of satellite images on available machine learning methods are the key to gathering reliable information for LULC management on different scales. On the other hand, there are many classification algorithms available in the literature for LULC mapping [19], from low computational costs such as machine learning [20] to deep learning [20,21]. Among them, the Random Forest (RF) algorithm [22] has been widely used in the context of LULC classification methods, favored for its robustness, the relatively low cost of the computational process, and because it does not require many parameters and data to obtain high-accuracy results [23]. RF excels at handling complex datasets, mitigating overfitting with higher accuracy compared to other methods (e.g., Support Vector Machines and Artificial Neural Networks [24]), and allows the use of a variety of datasets (e.g., vegetation indexes, fraction images, and spectral bands) to improve LULC classification. This algorithm does not consider a priori statistic distribution and it also combines a set of features randomly to create trees with bootstrapped samples of the training data [25], also presenting the estimated importance of the variables for classification [26–28]. Then, many decision trees are generated, and an unweighted selection is used to assign an unknown pixel to a specific class. With the advance of cloud computing platforms like Google Earth Engine (GEE) large datasets can be processed [13,29], especially with the addition of algorithms such as RF in the platform. For instance, large-scale data at state, country, or continental levels can have great results after a short processing time to combine the multiple datasets available on the GEE platform [30].

The Brazilian Annual Land Use and Land Cover (LULC) Mapping Project (Map-Biomas) initiative was formed in 2015 to develop an annual time series of the Brazil's LULC maps from 1985 to present with a spatial resolution of 30 m. The project is organized by biomes (such as the Amazon, Atlantic Forest, Caatinga, Cerrado, Pampa, and Pantanal) and cross-cutting themes (such as pasture, agriculture, forest plantation, coastal zone, mining, and urban area). The imagery dataset used in the MapBiomas project for collections 1 to 7 is composed by the Landsat Thematic Mapper (TM), Enhanced Thematic Mapper Plus (ETM+), and the Operational Land Imager and Ther-

mal Infrared Sensor (OLI-TIRS) sensors onboard of Landsat 5, Landsat 7 and Landsat 8, respectively, due to the long-term data available at medium spatial resolution (https://mapbiomas-br-site.s3.amazonaws.com/ATBD_Collection_7_v2.pdf, accessed on 10 June 2023). Despite MapBiomas being widely used in Brazil, its complex data preprocessing and algorithms may hinder its full reproducibility in other areas. In contrast, the proposed method introduces a novel approach that enhances reproducibility and accuracy. By selecting images based on the spectral and temporal characteristics of the LULC classes and sorting them according to complexity, from the most uniform to the most complex, the classification process becomes more systematic and controlled. Classifying one LULC class at a time in that order allows for a more focused and accurate classification, reducing potential errors and ensuring better mapping results. This method promotes transparency and replicability, making it easier for researchers to apply the same approach to different regions with similar characteristics.

Therefore, this work aims to map the LULC in the state of São Paulo using the Landsat-8 Operational Land Imager (OLI) images acquired mainly during the year 2020, based on the classification of individual LULC classes: forest, forest plantation, pasture, agriculture, urban areas, and water. For forest plantation exclusively we also used images from Landsat-8 OLI from the years 2013 to 2020 due to its rotation age [31]. First, we identified the individual classes and then we highlighted these classes by deriving synthetic images such as spectral indices and fraction images. Finally, we applied the RF algorithm, available on the GEE platform, to classify each LULC class separately, using the best set of visually selected images, and then compose the LULC map of São Paulo State.

## 2. Materials and Methods

### 2.1. Study Area

The state of São Paulo (Figure 1) is located in the Southeast Region of Brazil, with an approximate area of 248,209 km$^2$, between the parallels 19°43′18″ and 25°22′55″ south and the meridians 44°7′38″ and 53°8′10″ west. This study area was chosen because it presents a wide variety of LULC classes with spatial, spectral, and temporal variability, resulting in a landscape with great complexity. The vegetation types observed in the São Paulo State are dense ombrophilous forest, mixed ombrophilous forest, semi-decidual seasonal forest, restinga, savanna, and mangrove. They vary accordingly to climate, soil, relief, and distance from the Atlantic coast, and also have mosaics of occupation and anthropic uses [32].

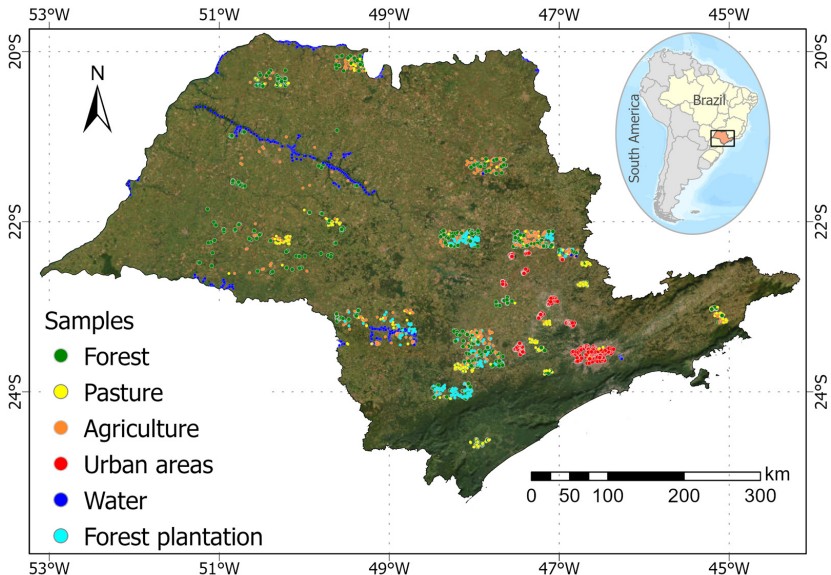

**Figure 1.** Geographical location of São Paulo State (Brazil, South America) and the Google Earth image showing the samples used to validate the LULC classification.

Much of São Paulo State's original vegetation cover was intensely devastated by logging, the economic cycles of coffee, sugarcane, orange, cattle, and urban growth. In the last survey of vegetation cover (2020), 5.6 million hectares of native vegetation were mapped, representing 22.9% of the territory [16]. Currently, approximately 70% of the São Paulo State's territory is occupied by agriculture (pasture and crops) [11], emphasizing that, economically, São Paulo is the most important state for sugarcane and orange cultivation in Brazil, corresponding to 55.1% and 63.9%, respectively, in relation to the national total planted area [33,34].

### 2.2. Remote Sensing Data

The multi-temporal Landsat 8 OLI images, converted to top-of-atmosphere (TOA) reflectance by USGS (Product: Landsat Surface Reflectance) [35] were used to develop the proposed method. From the original spectral bands, we generated fraction images derived from the Linear Spectral Mixing Model (LSMM) [36] and spectral indices to highlight the LULC classes that were used as input variables in the classification process. In addition, we used the Shuttle Radar Topography Mission (SRTM) data to help the classification [37,38].

The period defined for generating the image mosaics was based on the characteristics of the LULC classes and the lower presence of clouds (<40%) in the image, for example, a unique mosaic from January to June 2020, as shown in Figure 2. Different periods were used in the classification to consider the characteristics of the LULC classes. For water classification an annual mosaic showing the water extent in 2020 was used; for urban and forest the period was from May to September (dry season); and for agriculture from January to June 2020 (the growth season). We also used images from Landsat-8 OLI from the May to September of 2013 to 2020 time period to detect longer rotation cycles to classify forest plantation (most *Eucalyptus* plantations have around a 7-year rotation cycle).

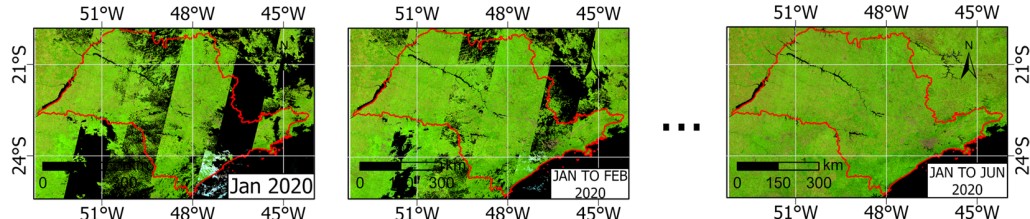

**Figure 2.** Example of monthly image mosaics and a cloudless image mosaic generated for the period analyzed (January to June 2020) from Landsat-8 OLI images (RGB color composite) over São Paulo State.

### 2.2.1. Spectral Indices

As described in Table 1, we have considered six spectral indices during the classification procedure.

**Table 1.** Spectral indices used to classify LULC and their equations based on Landsat spectral bands.

| Index | Equation | Reference |
|---|---|---|
| Normalized Difference Vegetation Index (NDVI) | $\frac{NIR - Red}{NIR + Red}$ | [39] |
| Enhanced Vegetation Index (EVI) | $2.5 \frac{NIR - Red}{NIR + 6 \times Red - 7.5 \times Blue + 1}$ | [40] |
| Green Normalized Difference Vegetation Index (GNDVI) | $\frac{NIR - Green}{NIR + Green}$ | [41] |
| Normalized Difference Water Index (NDWI) | $\frac{NIR - SWIR1}{NIR + SWIR1}$ | [42] |
| Normalized Burn Ratio (NBR) | $\frac{NIR - SWIR2}{NIR + SWIR2}$ | [43] |
| Normalized Difference Urban Index (NDUI) | $\frac{NTL - NDVI}{NTL + NDVI}$ | [44] |

NTL = normalized Operational Linescan System (OLS) on the Defense Meteorological Satellite Program (DMSP) nighttime light data.

### 2.2.2. Fraction Images

To obtain fraction images, we applied the LSMM method [36], which assumes that pixel values are linear combinations of reflectance from the pixel's components, called endmembers:

$$R_i = \sum_{j=i}^{n} f_i r_{i,j} + \varepsilon_i \tag{1}$$

where $R_i$ represents the spectral reflectance in the $i$th spectral band; $r_{i,j}$ is the spectral reflectance of the $j$th component in $i$th spectral band (endmember); $f_i$ is the proportion of the $j$th component within the pixel; and $\varepsilon_i$ is the residual for the $i$th spectral band.

The spectral endmembers (vegetation, soil, and shade/water) were selected by analyzing the spectral and temporal characteristics of these endmembers in the images (endmember value extracted directly from the images analyzed). The vegetation endmember was selected in areas of dense vegetation cover, the soil endmember was selected in areas of exposed soil without vegetation cover, and the shade endmember was selected in areas occupied by water since shade and water have similar lower spectral responses than the other two endmembers considered in this study [45]. Then, vegetation, soil, and shade fraction image mosaics were generated, discriminating from the abundance of each endmember inside the pixels.

### 2.3. Methodological Approach

The classes of forest, forest plantation, pasture, agriculture, urban areas, and water were classified according to their complexity when analyzed using Landsat 8 OLI images. In this way, first we classified water, followed by urban area, forest, agriculture, and lastly forest plantation. We assumed that all remaining areas were covered by pasture. This is shown in the flowchart of the methodological approach used to classify the LULC classes in the state of São Paulo for the year 2020 (Figure 3).

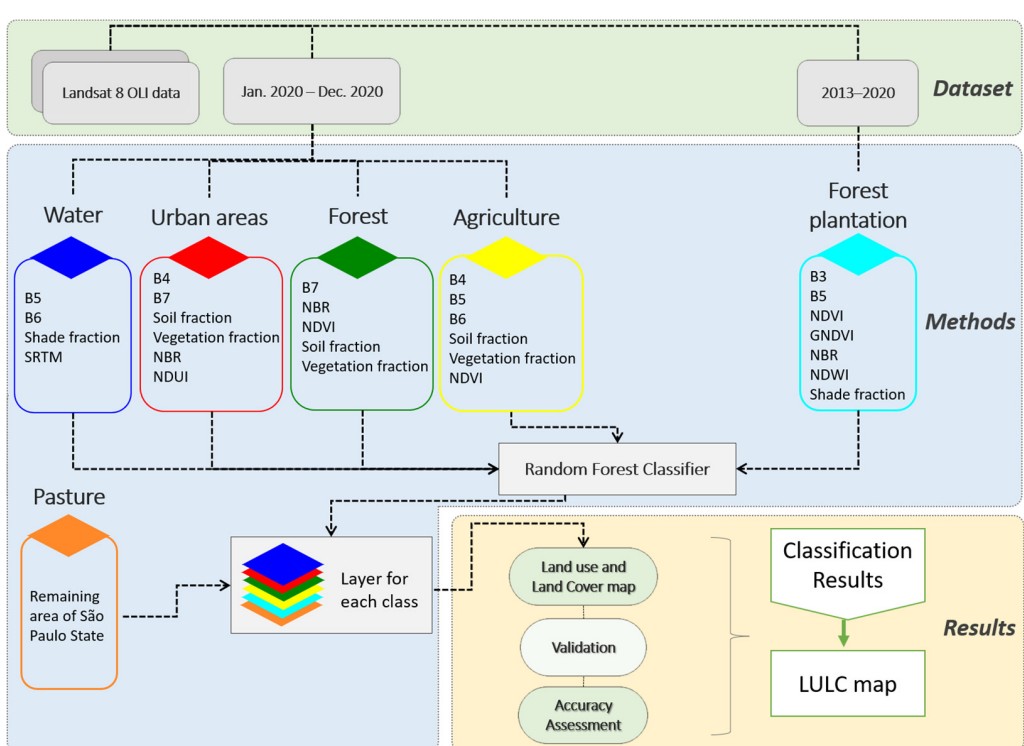

**Figure 3.** Flowchart of the methodological approach.

### 2.3.1. Landsat-8 OLI Image Analysis

For this study, we used a mosaic for the January to June 2020 period (Figure 4A) composed of 379 images, mostly comprising the end of wet season and the beginning of

dry season and another for the July to September 2020 period (Figure 4B) composed of 212 images. The first image mosaic shows the green vegetation (wet season) and the second one shows a higher number of bare soil areas (dry season).

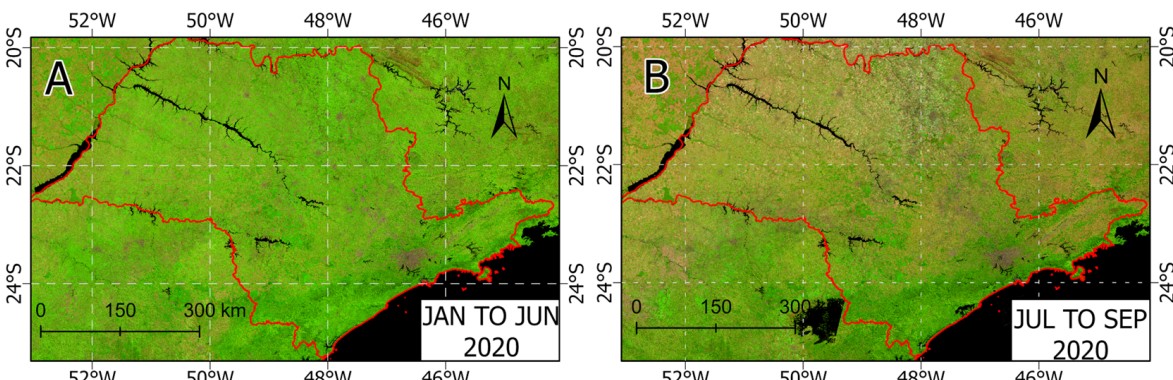

**Figure 4.** The image cloudless mosaics used in this work: (**A**) January to June 2020 and (**B**) July to September 2020 of Landsat-8 OLI sensor data (RGB color composites) of the São Paulo State.

Based on the mosaics, we investigated which time period was more suitable for the classification of each class and found that an annual mosaic (January to December 2020) improves the water classification; for the urban and forest classes, the better classification results were achieved based on the period from May to September 2020 (the dry season); for the agriculture classification, we used mosaic images from January to June (the growing season), that highlighted this class area. We also used images from Landsat-8 OLI from May to September of the period from 2013 to 2020 to detect longer rotation cycles to classify forest plantations (most *Eucalyptus* plantations with a rotation cycle of approximately 7 years). Then, we generated several derived images for each class period such as vegetation indices (Table 1), fraction images (vegetation, soil, and shade/water), and some statistical information (e.g., ratio and percentiles). The LSMM was applied to each image and then we calculated the maximum and standard deviation of the fractions. The LSMM was also applied in the medians of the two periods. The Normalized Difference Urban Index (NDUI), for example, was applied to the image mosaics to highlight the urban areas (Figure 5).

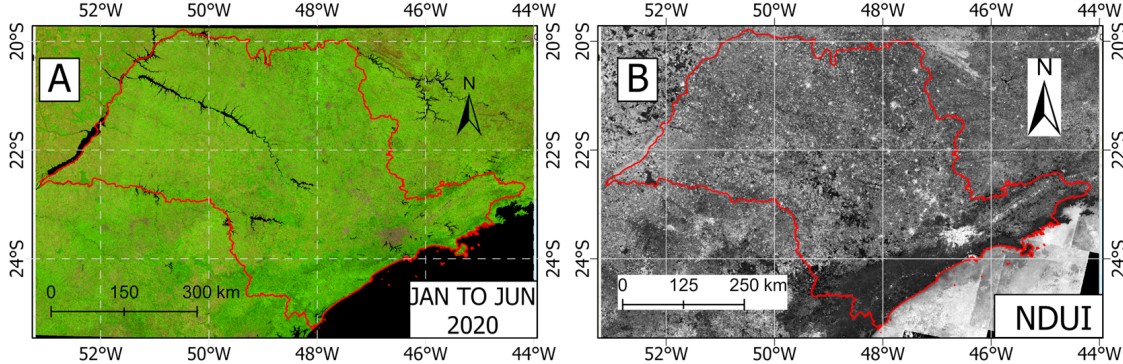

**Figure 5.** (**A**) the corresponding image mosaic from January to June 2020 of Landsat-8 OLI sensor data (RGB color composite), and (**B**) The NDUI highlighting the urban areas (in white color) for the image mosaic from January to June 2020, over São Paulo State.

### 2.3.2. Image Classification

To perform the image classification, we applied the RF classifier algorithm. RF requires a set of parameters, such as the number of trees and subsampling, that influence the classification results. The classifications were performed individually for each LULC class. All the data and input variables used to classify each LULC using the RF classification are

described in (Table 2). We considered each data used for the RF classification according to the spectral response of each LULC target, therefore we had previously selected the OLI bands and generated spectral indices to filter the input data for RF classification.

**Table 2.** Input variables used for classifying the LULC classes in the study area.

| LULC Classes | Data | Input Variables |
|---|---|---|
| Water | B5 | Percentiles 10, 25, 50, 75, 90; |
| | B6 | Percentiles 10, 25, 50, 75, 90; |
| | Shade fraction | Percentiles 10, 25, 50, 75, 90; |
| Urban areas | B4 | Percentiles 10, 25, 50, 75, 90; |
| | B7 | Percentiles 10, 25, 50; |
| | Soil fraction | Percentiles 10, 25, 50, 75, 90; |
| | Vegetation fraction | Percentiles 10, 25, 50, 75, 90; |
| | NDUI | Percentiles 50, 75, 90; |
| Forest | B7 | Percentiles 50, 90; |
| | NBR | Percentiles 50, 75; |
| | NDVI | Percentiles 50, 90; |
| | Soil fraction | Percentiles 25, 50; |
| | Vegetation fraction | Percentiles 50, 75, 90; |
| Forest plantation | B3 | Percentiles 25, 50; |
| | B5 | Percentile 90; |
| | NDVI | Percentiles 50, 90; |
| | GNDVI | Percentiles 50, 75, 90; |
| | NBR | Percentiles 25, 75, 90; |
| | NDWI | Percentile 50; |
| | Shade fraction | Percentiles 10; |
| | Vegetation fraction | Percentiles 50, 75, 90; |
| Agriculture | B4 | Percentiles 10, 25, 50, 75, 90; |
| | B5 | Percentiles 10, 25, 50, 75, 90; |
| | B6 | Percentiles 10, 25, 50, 75, 90; |
| | Soil fraction | Percentiles 10, 25, 50, 75, 90, and standard deviation; |
| | Vegetation fraction | Percentiles 10, 25, 50, 75, 90, and standard deviation; |
| | NDVI | Percentiles 10, 25, 50, 75, 90; |

Initially, the classification for the water was performed because it is considered the easiest class to identify compared to the other classes in this study. However, some challenges were identified when analyzing the results such as, for example, the shade of the slopes were misclassified as water. To minimize this effect, SRTM data [37,38] was used to calculate the slope and to exclude those areas classified as water with a slope greater than 17%. The input variables selected to classify the water were the percentiles of shade fraction and spectral bands 5 (NIR) and 6 (SWIR 1) of the January–June image mosaic and SRTM.

For the urban areas classification, the percentiles of vegetation and soil fraction images, spectral bands 4 (red) and 7 (SWIR 2), NDUI, and NBR were selected. Also, we used night-time lights information obtained from VIIRS/DNB (Visible Infrared Imaging Radiometer Suite/Day&Night band) [46] as ancillary data to highlight the urban areas.

For the agriculture areas classification, we selected spectral bands 4 (red), 5 (NIR), and 6 (SWIR), NDVI, vegetation and soil fractions, maximum value and standard deviations of the vegetation and soil fractions from the January–June time period considered as the rainy season when the main crops grow. We considered the initial classification for agriculture in the most-planted cultures in São Paulo State using the following works as bases: soybeans and annual cultures [47]; coffee and citrus [48]; and sugarcane [49]. They represent most of São Paulo's agriculture area. Then, we compiled all the crops in the agriculture class.

For the forest areas classification, we selected the spectral band 7 (SWIR 2), NDVI, NBR, vegetation and soil fraction images from the July–September time period considered as the dry season [50]. For the forest plantation areas, the classification method was based on a previous study using time-series mosaic products from 2013 to 2020 [31]. The images selected for classification were the shade and vegetation fractions, spectral band 3 (green), NDVI, GNDVI, NBR, and NDWI of annual image mosaics. We used the temporal series of images because it allows us to capture the cutting cycle of the main cultivated species, with an emphasis on *Eucalyptus*. After those classifications, we added the pasture class. It was classified as the remaining area of São Paulo State, based on the difference in mosaicking from the previous LULC classes.

For the RF algorithm, we set parameters for each class (N° of trees, nodes, subsampling) (Table 3). We set 101 trees for each classification as we were testing the best number for the classification, and 50 trees for the agriculture class. For all classes we used a sampling factor of 0.8. Then, after running the classification the number of nodes for each class and the out-of-bag error estimate (OOB error) were presented. The OOB error evaluates the misclassification error unbiasedly, considering the bootstrap out-of-bag samples [22,24,51].

**Table 3.** Random Forest parameters used for each class.

| Classes | Water | Urban Areas | Forest | Agriculture | Forest Plantation |
|---|---|---|---|---|---|
| N° of trees | 101 | 101 | 101 | 50 | 101 |
| Subsampling | 0.8 | 0.8 | 0.8 | 0.8 | 0.8 |
| N° of nodes | 1320 | 2849 | 10684 | 21,228 | 8843 |
| out-of-bag error estimate | 0.001 | 0.045 | 0.007 | 0.032 | 0.002 |

### 2.4. Validation of the Classification

To validate the results, we created 500 random polygons for each LULC class based on visual interpretation of Sentinel-2 images for the year 2020 across the São Paulo State. We selected images from January, February, March, April, November, and December. To ensure that the classes had no changes, we validated the collected samples in December. Also, time series of EVI and NDVI indices were used to visualize the spectral patterns of each class and support the decision-making. We selected the centroid of each polygon to extract the class and to perform a confusion matrix for cross-validation [52]. For each class, we collected the samples based on the heterogeneity of the classes according to the visual interpretation described in Table 4 and shown in Figure 6.

We also compared our classification results with the MapBiomas collection 6.0 LULC map for the year 2020 [15]. The project generated 35 annual LULC maps for Brazil processed using the GEE with a three-level classification legend [15]. The first level of the classification legend is primarily focused on distinguishing land cover vegetation types (e.g., vegetation, farmlands, non-vegetated areas, water, etc.). The second level distinguishes between natural forest, newly planted forest, grasslands, wetlands, urban infrastructure, plantations, ranchlands, and livestock and agricultural farms. The third level was specifically focused on forest formation [15]. Based on the levels of classification legend, the MapBiomas LULC product was reclassified and grouped into classes similar to the classification performed by our method. Then, the kappa value was calculated to verify the agreement of those classification results.

**Table 4.** Description of the criteria used to collect the samples in the Sentinel-2 images to validate the LULC classification.

| LULC | Criteria Used to Collect the Samples |
|---|---|
| Water | We used the NIR-SWIR-Red to highlight the water in dark shades and colors. |
| Urban areas | We used the RGB natural composition to maintain a spatial resolution of 10 m and a mosaic of high-resolution images from Google Earth was also used to confirm the collected samples. |
| Forest | We selected the NIR-SWIR-Red composition to enhance the textures commonly used to identify this class by visual interpretation. Its rough texture and no changes in the time allow its identification on satellite images. |
| Agriculture | The agriculture samples had an analysis of the crop's development, ensuring the class's assertiveness. We selected the NIR-SWIR-Red composition, to characterize the spectral response of crops along the rotation cycle of soybean, corn, bean, and sugarcane. |
| Forest plantation | We selected the NIR-SWIR-Red composition to enhance the textures commonly used to identify this class by visual interpretation. This class presents smooth texture, long cycles, and bright spectral response for recently planted forestry. |
| Pasture | We used the RGB natural composition, given the spectral response of the pasture in light green tones. In addition, due to the spatial resolution of 10 m, it is also possible to identify the paths taken by the cattle. |

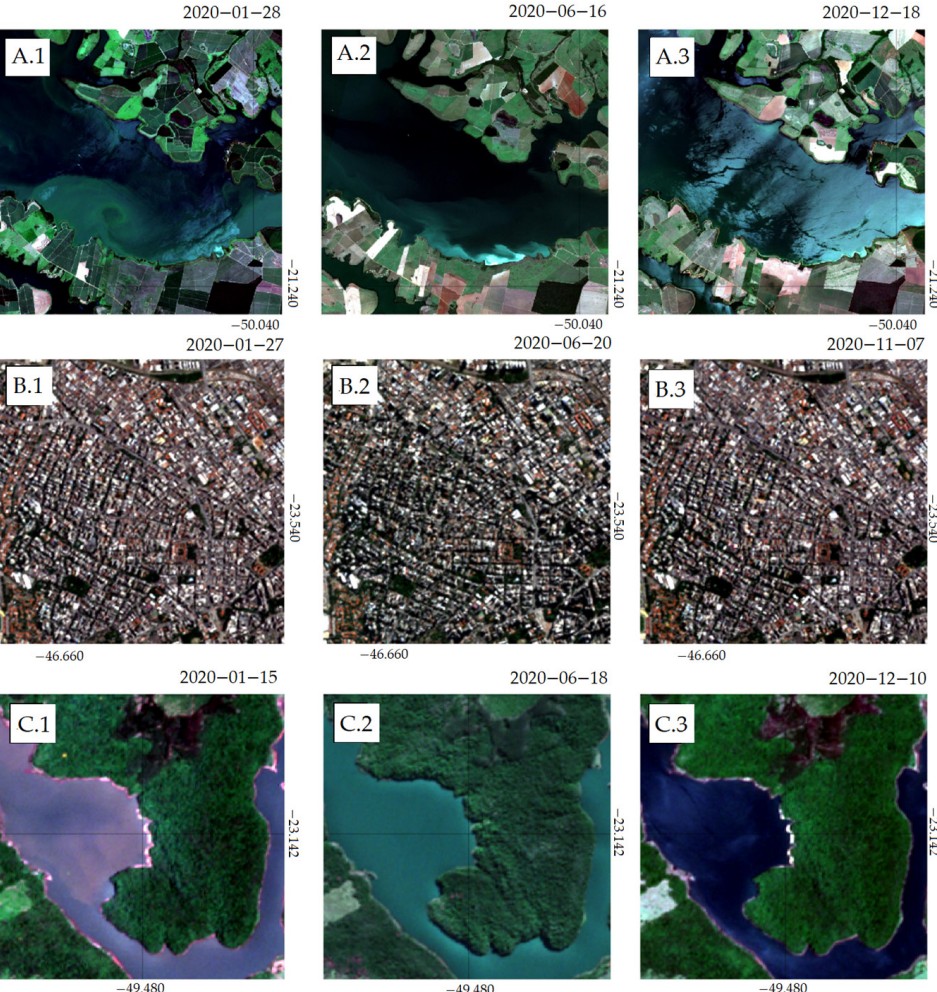

**Figure 6.** *Cont.*

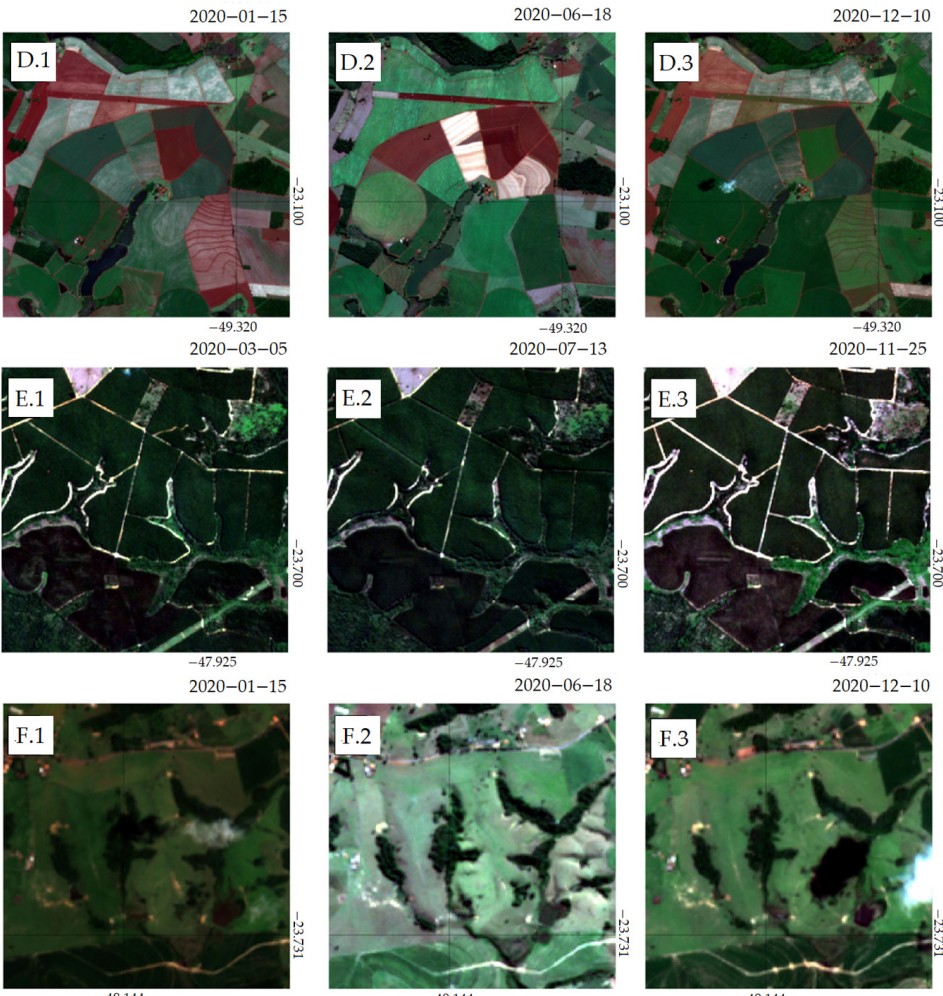

**Figure 6.** Sentinel-2 images of the samples collected to validate each class in true color composition: (**A.1**–**A.3**). Water; (**B.1**–**B.3**). Urban area; (**C.1**–**C.3**). Forest; (**D.1**–**D.3**). Agriculture; (**E.1**–**E.3**). Forest Plantation; (**F.1**–**F.3**). Pasture. Three images for each LULC class acquired during 2020 year were analyzed to select the samples used for the classification evaluation.

## 3. Results

Figure 7 shows the LULC map of São Paulo State for the year 2020 produced by the method proposed in this work. We observed that most of the forest areas are in the eastern region of São Paulo, mostly linked to the steeper slopes. It covers 47,056 km$^2$ (18.96%), and most of it is in the Atlantic Rain Forest biome. Only small and sparse patches of Cerrado forests still remain as natural vegetation, mostly located in protected areas. Pasture is the class with the largest proportion of the study area, representing 96,808 km$^2$ (39%) of the total area. Pastures are found throughout the state, but the east and west are the two most representative regions. The agriculture class comprised 85,243 km$^2$ (34.34%), located around the Tiete river (the big river at the center), and the southern and central northern parts of São Paulo State. Forest plantation covers 9034 km$^2$ (3.64%) and is particularly located in the central south region. The São Paulo metropolitan area is the largest urban area in South America and the state has 4785 km$^2$ (1.93%) of its area classified as urban. Almost the same amount (5277 km$^2$) is covered by water, representing 2.12% of the state's area.

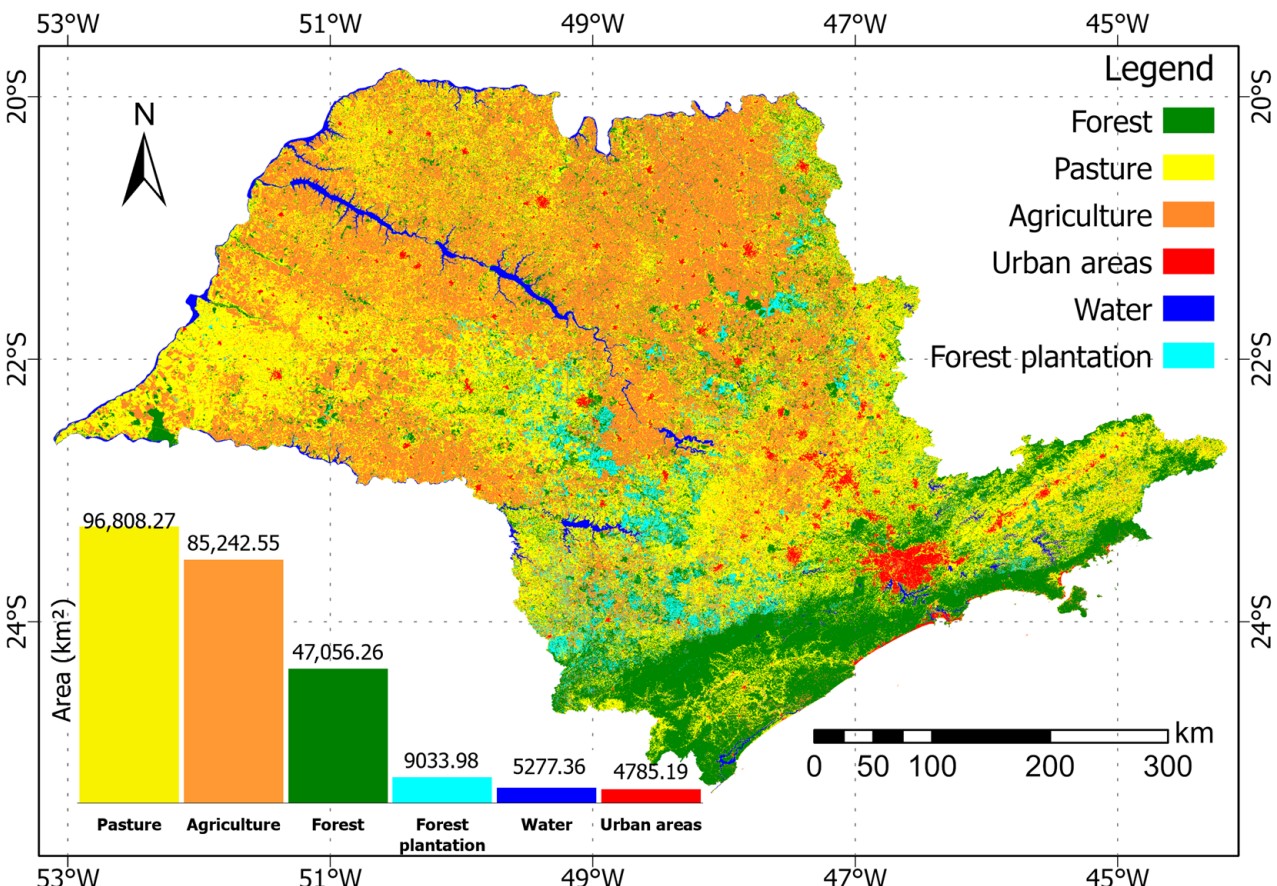

**Figure 7.** Land Use and Land Cover map of the São Paulo State for the year 2020. The classification was based on Landsat-8 OLI sensor data.

The importance variables obtained from the RF algorithm can be used to identify which variables were most important for classifying each class after our visual pre-selection. For the water class, band 6 (SWIR), shade fraction, and band 5 (NIR) in specific percentiles represented the three most important variables for RF with the input data used (Figure 8). The urban area class's three most important variables were based on image fractions (vegetation and soil) and spectral index (NDUI). The forest class had as its most important input the NDVI, vegetation fraction, and NBR index. For the forest plantation class, the spectral indices (GNDVI, NBR, and NDVI) presented a higher importance. The importance variables for the agriculture class were not obtained because the class was classified by a combination of cultures. Also, there are no importance variables for pasture since it was considered the remaining area of São Paulo State after classifying the other LULCs.

Table 5 shows the confusion matrix of the LULC classification, based on the collected samples, to evaluate the accuracy of the classification (prediction). The user and producer accuracies for each LULC class varied from 77% to 100% and 84% to 98%, respectively. The kappa value was 0.891 and the overall accuracy was 89.10%. The agriculture class presented the highest omission error since it is often misclassified as pasture and forest plantation due to the spectral similarities of these classes depending on the phenological response, and their spatial attributes (e.g., landscape fragmentation). On the other hand, the highest commission error was presented in the pasture class, which is often misclassified with other classes due to its heterogeneity, and temporal and spatial complexities.

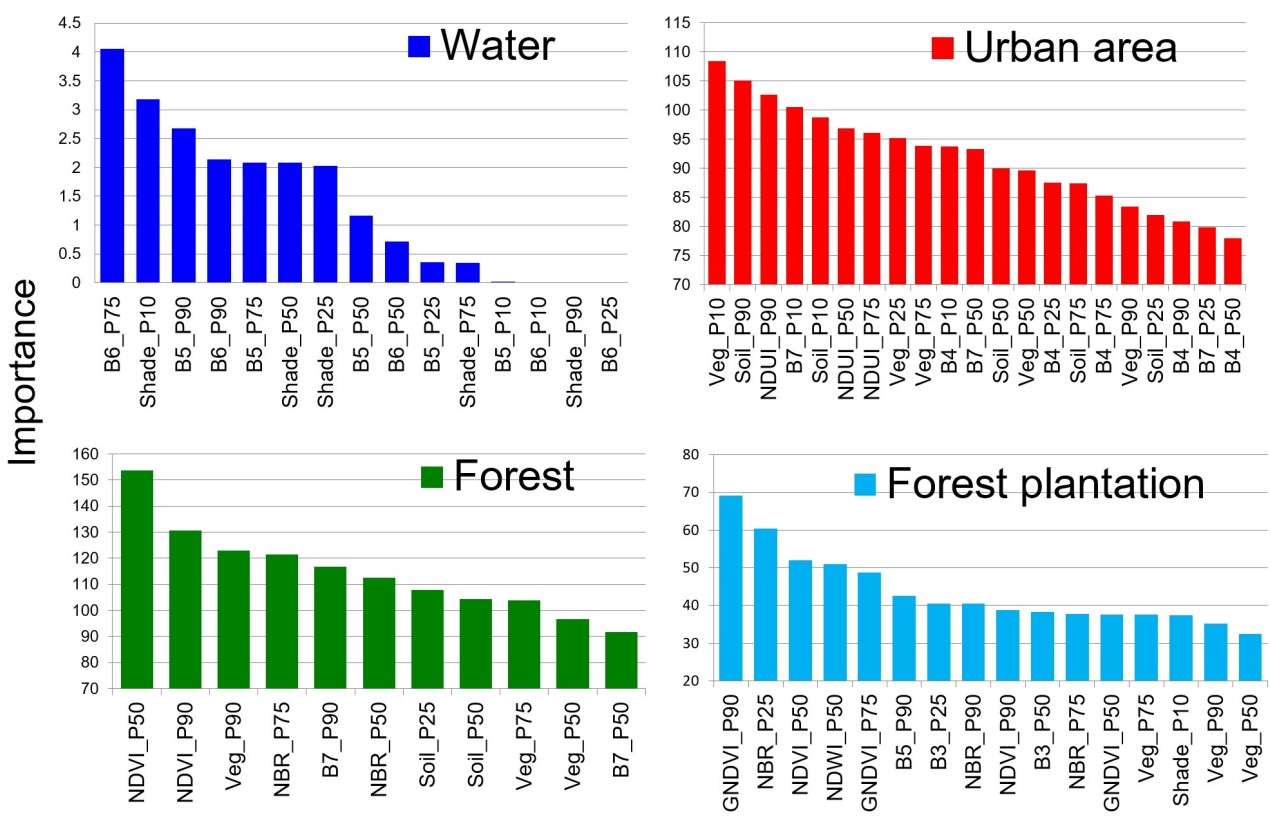

**Figure 8.** Importance variables for each class considered for the Random Forest classification algorithm.

**Table 5.** Confusion matrix for the LULC classification in 2020 (prediction) and collected samples (reference).

| | Reference | | | | | | |
|---|---|---|---|---|---|---|---|
| | **Forest** | **Pasture** | **Agriculture** | **Urban Areas** | **Water** | **Forest Plantation** | **User's Accuracy** |
| Forest | 446 | 36 | 9 | 1 | 3 | 35 | 84.15% |
| Pasture | 51 | 421 | 52 | 0 | 20 | 4 | 76.82% |
| Agriculture | 2 | 43 | 399 | 9 | 5 | 16 | 84.18% |
| Urban Areas | 0 | 0 | 0 | 490 | 0 | 0 | 100.00% |
| Water | 0 | 0 | 0 | 0 | 472 | 0 | 100.00% |
| Forest Plantation | 1 | 0 | 40 | 0 | 0 | 445 | 91.56% |
| Total | 500 | 500 | 500 | 500 | 500 | 500 | |
| Producer's Accuracy | 89.20% | 84.20% | 79.80% | 98.00% | 94.40% | 89.00% | |
| Kappa = | | 0.8692 | | Overall accuracy = | | 89.10% | |

The confusion matrix for the MapBiomas LULC classification was cross-validated with the same collected samples shown in Table 6. Although this data is already published and has an overall accuracy for level 2 data of 87.4% [15,18], we used the same collected samples to validate and resemble our results with the same reference data. The user and producer accuracies for each mapped class varied from 78% to 100% and 79% to 100%, respectively. The kappa value was 0.926 and the overall accuracy was 93.90%, based on the validation

using our collected samples. For this data, the highest errors are in the agriculture and pasture classes (Table 6).

**Table 6.** Confusion matrix for the MapBiomas LULC classification in 2020 (prediction) and collected samples (reference).

| | Reference | | | | | | |
| | Forest | Pasture | Agriculture | Urban Areas | Water | Forest Plantation | User's Accuracy |
|---|---|---|---|---|---|---|---|
| Forest | 492 | 0 | 0 | 1 | 4 | 2 | 98.60% |
| Pasture | 7 | 394 | 34 | 0 | 1 | 3 | 89.75% |
| Agriculture | 1 | 106 | 464 | 0 | 0 | 21 | 78.38% |
| Urban Areas | 0 | 0 | 2 | 499 | 1 | 0 | 99.40% |
| Water | 0 | 0 | 0 | 0 | 494 | 0 | 100.00% |
| Forest Plantation | 0 | 0 | 0 | 0 | 0 | 474 | 100.00% |
| Total | 500 | 500 | 500 | 500 | 500 | 500 | |
| Producer's Accuracy | 98.40% | 78.80% | 92.80% | 99.80% | 98.80% | 94.80% | |
| Kappa = | 0.9268 | | | Overall accuracy = | | 93.90% | |

In this manner, we performed the confusion matrix using the same sample points to evaluate the agreement of our result with the MapBiomas product that uses more a complex method (Table 7). The user and producer accuracies for each mapped class varied from 62% to 100% and 63% to 98%, respectively. The kappa value was 0.83 and the overall accuracy was 85.47% showing an excellent agreement [52]. Again, the agriculture class presented the highest omission error, and was often misclassified as pasture, forest plantation, and forest. For the commission error here, the pasture also presented the highest error.

**Table 7.** Confusion matrix for the LULC classification in 2020 (prediction) and MapBiomas (reference).

| | Reference (MapBiomas) | | | | | | |
| | Forest | Pasture | Agriculture | Urban Areas | Water | Forest Plantation | User's Accuracy |
|---|---|---|---|---|---|---|---|
| Forest | 445 | 29 | 41 | 0 | 3 | 12 | 83.96% |
| Pasture | 52 | 340 | 139 | 1 | 15 | 1 | 62.04% |
| Agriculture | 1 | 69 | 373 | 10 | 5 | 16 | 78.69% |
| Urban Areas | 0 | 0 | 0 | 490 | 0 | 0 | 100.00% |
| Water | 0 | 0 | 0 | 1 | 471 | 0 | 99.79% |
| Forest Plantation | 1 | 1 | 39 | 0 | 0 | 445 | 91.56% |
| Total | 499 | 439 | 592 | 502 | 494 | 474 | |
| Producer's Accuracy | 89.18% | 77.45% | 63.01% | 97.61% | 95.34% | 93.88% | |
| Kappa = | 0.8257 | | | Overall accuracy = | | 85.47% | |

## 4. Discussion

The aim of this study was to generate a LULC map for the state of São Paulo for the year 2020, based on 13 variables and their respective attributes derived from remote sensing indices, fraction images, and spectral bands used in RF models. The novelty of the

proposed method is the performance of a preliminary analysis to select the features that enhance the individual LULC classes in the OLI images. In addition, it defines the period of production of the cloud-free image mosaics to be processed. Then, the LULC classes were classified individually using specific spectral bands, spectral indices, and fraction images. Another factor that was considered was the temporal resolution. For example, for agriculture classification the image mosaic was composed of images acquired during the rainy season; for forest and forest plantation classifications, the image mosaic was composed of images acquired during the dry season. Multiannual mosaics were also used for the forest plantation class, taking into account the forest rotation age; we used mosaics from 2013 to 2020 [31]. These pre-processing steps highlighted the LULC classes for the classification step. In this study we used an RF classifier that is available in the GEE platform. RF is commonly used for LULC classification, due to its efficiency in achieving higher accuracy compared to other algorithms, and as the cost of the computational process is relatively low and does not require many parameters or much data to obtain high-accuracy results [23,53].

Classification errors between the agriculture and pasture classes occur mainly in areas of managed pasture. The rural producers who aim to increase the amount of food for cattle invest in pasture management destined for animal feeding [54,55]. In this sense, the vegetation indices and spectral responses from the satellite images could be similar between pasture and agricultural areas, with a significant gain in vegetation vigor at the beginning of the rains and a sudden loss of vegetation vigor at the end of the rainy season. For the selection of training points for the agriculture class, we selected images that consider the agriculture patterns based on the development of crops throughout the time series. In São Paulo State, the main crops are soybeans, sugarcane, corn, and beans [15,48,49].

Classification errors between agriculture and forest plantation classes occurred, in most cases, in areas with teak (*Tectona grandis*) and rubber tree (*Hevea brasiliensis*) plantations. These species lose their leaves during the dry period leading to a drop in the vegetation indices and consequently the spectral response is influenced by the soil spectral mixture, which can lead to confusion in the classifier. The RF classifier interprets this behavior as the beginning and end of a cycle, but it is the seasonality of the vegetation itself. The reference dataset used to assess the mapping accuracy was collected based on the visual interpretation of Sentinel-2 color composites and ancillary spectral indices.

The results obtained by the proposed method were compared to the existing LULC products from MapBiomas, which performs a more complex classification procedure [15,18]. In general, MapBiomas classification presented better results (Table 6) compared to our method (Table 5). However, our proposed method presented lower omission error for the pasture class (84%) compared to MapBiomas (79%) and lower inclusion error for the agriculture class (84%) in comparison (78%). The differences observed between the results of the proposed method and the MapBiomas project can be attributed to several factors, for example, the classification algorithm, the variables and training dataset used in the classification process, and the data processing are some of the main factors that impact the classification results between the LULC maps analyzed. Our proposed method was trained and applied specifically for the state of São Paulo and achieved an overall accuracy of 89% based on the 13 variables and their respective attributes used in the classification. The pre-selection of variables can reduce data volume without losing important information, supporting the discrimination of classes and also improving classification accuracy. This study also analyzed the importance of the variables in the RF models, however, further investigation is necessary to understand the characteristics of each class of LULC in the study area.

## 5. Conclusions

In this study we selected the images from sensors onboard Landsat satellites to classify each class individually and produce a LULC map for the state of São Paulo, located in the Southeast Region of Brazil. For all classes, except forest plantation, we used images

acquired during 2020. For forest plantations, we used images acquired over a long period (2013–2020) due to its rotation growth. The pre-processing performed highlights the LULC classes prior to the classification step. The proposed method facilitates post-classification editing for each class individually to ensure the high accuracy of the map. The validation process showed that the user and producer accuracies for each mapped class varied from 76.62% to 100% and 84.20% to 98%, respectively, the kappa value was 0.8692, and the overall accuracy was 89.10%. The comparison with the MapBiomas LULC map for the same year from collection 6.0, which uses a more complex classification process, showed an agreement of a kappa value equal to 0.8257 and an overall accuracy of 85.47%. The advantage of our method is that it is more flexible to apply for any region and any time period with different LULC classes.

**Author Contributions:** Conceptualization, Y.E.S., E.A., V.D. and P.R.M.; methodology, Y.E.S., E.A., G.M.d.S., T.B.H., G.M., A.C.D., M.A. and H.L.G.C.; writing—original draft preparation, Y.E.S., E.A., G.M.d.S., G.M. and H.L.G.C.; writing—review and editing, Y.E.S., E.A., G.M.d.S., T.B.H., H.L.G.C., A.C.D., V.D., P.R.M., M.A. and G.M. All authors have read and agreed to the published version of the manuscript.

**Funding:** This work was funded by the São Paulo Research Foundation (FAPESP-grant 2019/19371-5. It was supported by the Brazilian National Council for Scientific and Technological Development (CNPq-grant 303299/2018-5) and the Improvement of Higher Education Personnel (CAPES—financing code 001). H.L.G.C thanks the FAPESP grant 18/14423-4. G.M. thanks the FAPESP grants 2019/25701-8 and 2023/03206-0. A.C.D. thanks the FAPESP grants 2022/01746-5 and 2023/02386-5. M.A. thanks the CNPq grant (PQ—306334/2020-8).

**Data Availability Statement:** All data and code used in the current study are available from the corresponding author on reasonable request.

**Acknowledgments:** The authors are grateful to Instituto Nacional de Pesquisas Espaciais (INPE) for research support. Also, the authors would like to thank the handling editors and the anonymous reviewers for their valuable comments and suggestions, which significantly improved the quality of this paper.

**Conflicts of Interest:** The authors declare no conflict of interest.

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
