# Peer review of "Mapping Land Use and Land Cover Classes in São Paulo State, Southeast of Brazil, Using Landsat-8 OLI Multispectral Data and the Derived Spectral Indices and Fraction Images"

_forests, doi:10.3390/f14081669_

Round 1

Reviewer 1 Report (New Reviewer)

The manuscript proposes a methodology to map the LULC of Sao Paulo. To his aim, the authors use long-term Landsat data and the Random forest classification method. The paper gives the aim of the study and relevant steps of the methodology. However, there are some questions and suggestions that the authors should consider, given below;  

1.      The introduction part should include literature that uses MapBiomas.

2.      Why Sentinel-2 data which has better spatial, spectral, and temporal resolution than Landsat was not considered for the LULC mapping?

3.      Page 4 line 140: The abbreviation “LUCL” should be “LULC”.

4.      In Figure 5 put a legend for the NDUI map. Which color represents urban? It should be written.

5.      Page 8 lines 247-248: The sentence in the parenthesis should be removed, seems an error in the format.

6.      Numbers of the training data for each class should be provided.

7.      All figures should be recreated. Most words cannot be read because they look blurry.

8.   Page 12: there are two same figures.

9.   When the recommended method and the MapBiomas evaluated with the same reference data in the study are compared, it is seen that the MapBiomas data is better than the proposed approach. Table 7 shows the agreement between the two data. However, in the paper, the results indicated in Tables 5 and 7 are emphasized. The difference between the proposed data and MapBiomass should be given such as the data, method, approach, etc. The limitations of the study should be also provided, considering the complexity of the region as mentioned in the introduction.

10.   The discussion part should be expanded, considering the literature on MapBiomass data and applications. One example is given below, please check the literature for others.

https://www.mdpi.com/2072-4292/13/15/2853

There are some minor typos given below;

1.      Page 2 line 50: what is “ignition”? Is there a typing error?

2.      Page 4 line 128: The word “show” should be “shows”.

3.   The first sentence of the discussion should be revised. “…showed that it produced a ….”

Author Response

Reviewer-1

Comments and Suggestions for Authors

The manuscript proposes a methodology to map the LULC of Sao Paulo. To his aim, the authors use long-term Landsat data and the Random forest classification method. The paper gives the aim of the study and relevant steps of the methodology.

Authors: Thanks very much for the positive comments. All comments were considered and improved the revised manuscript.

However, there are some questions and suggestions that the authors should consider, given below;  

  1. The introduction part should include literature that uses MapBiomas.

Authors: Thanks for your suggestion. We have included some references of MapBiomas in the introduction and through the text in the revised manuscript.

  1. Why Sentinel-2 data which has better spatial, spectral, and temporal resolution than Landsat was not considered for the LULC mapping?

Authors: Thanks for your comment. You are right, Sentinel-2 data has better spatial, spectral, and temporal resolution than Landsat and were used to validate our results and MapBiomas results.

  1. Page 4 line 140: The abbreviation “LUCL” should be “LULC”.

Authors: Thanks for your correction. It was changed in the revised manuscript.

  1. In Figure 5 put a legend for the NDUI map. Which color represents urban? It should be written.

Authors: Thanks for your suggestion. The color (white) represents the urban areas. It was written in the revised manuscript.

  1. Page 8 lines 247-248: The sentence in the parenthesis should be removed, seems an error in the format.

Authors: Thanks for your suggestion. The sentence in the parenthesis was removed in the revised manuscript.

  1. Numbers of the training data for each class should be provided.

Authors: Thanks for your suggestion. Numbers of the training data for each class were included in the revised manuscript as suggested.

  1. All figures should be recreated. Most words cannot be read because they look blurry.

Authors: Thank you for your suggestion. The figures have been recreated.

  1. Page 12: there are two same figures.

Authors: Thanks for your comments. One figure was removed in the revised manuscript.

  1. When the recommended method and the MapBiomas evaluated with the same reference data in the study are compared, it is seen that the MapBiomas data is better than the proposed approach. Table 7 shows the agreement between the two data. However, in the paper, the results indicated in Tables 5 and 7 are emphasized. The difference between the proposed data and MapBiomass should be given such as the data, method, approach, etc. The limitations of the study should be also provided, considering the complexity of the region as mentioned in the introduction.

Authors: Thanks for your comments. You are right, the results of MapBiomas are better than our results as described in the revised manuscript. Then we compared our results with MapBiomas results to verify the agreement of both products. The comparison showed that there is an excellent agreement of both results as described in the revised manuscript.

  1. The discussion part should be expanded, considering the literature on MapBiomass data and applications. One example is given below, please check the literature for others.

https://www.mdpi.com/2072-4292/13/15/2853

 Authors: Thanks for your suggestion. This reference was included in the Introduction section of the revised manuscript.

There are some minor typos given below;

  1. Page 2 line 50: what is “ignition”? Is there a typing error?

Authors: Thanks for your observation. It was corrected in the revised manuscript.

  1. Page 4 line 128: The word “show” should be “shows”.

Authors: Thanks for your correction. It was changed in the revised manuscript.

  1. The first sentence of the discussion should be revised. “…showed that it produced a ….”

Authors: Thanks for your suggestion. It was changed in the revised manuscript.

Reviewer 2 Report (Previous Reviewer 1)

The effort of mapping LULC using Landsat-8, machine learning and spectral indices is such developing region, like Sao Paulo State is quite interesting and will bring a significant contribution in this field.

Beside this, manuscript needs serious improvements.

Some literature review of similar research should be added in introduction section. Novelty, in my opinion, should be also modified – current approach, presented in this manuscript, was applied in different studies (published in MDPI journals). Maybe authors should focus of “fraction images” as a modern approach.

L242-L249. What approach authors used to determine hyperparameters of the RF model for each class? Are they were selected manually, or authors used some ML tools? This part should also be clarified in this section.

Subsection 2.4. If authors used MapBiomas as a source of validation data – it should be described in this section.

Methods for validation of the RF model should also be described in Section 2

Figure 6, 7, 8 -  Image quality is too low, it will be a good idea to resave it in 300 dpi

I wish that my comment would be helpful in improving the quality of this research.

Thank you.

Author Response

Reviewer-2

Comments and Suggestions for Authors

The effort of mapping LULC using Landsat-8, machine learning and spectral indices is such developing region, like Sao Paulo State is quite interesting and will bring a significant contribution in this field.

Authors: Thanks very much for the positive comments. All comments were considered and improved the revised manuscript.

Beside this, manuscript needs serious improvements.

Some literature review of similar research should be added in introduction section. Novelty, in my opinion, should be also modified – current approach, presented in this manuscript, was applied in different studies (published in MDPI journals). Maybe authors should focus of “fraction images” as a modern approach.

Authors: Thanks for your comments and suggestions. We included additional references in the introduction section. We also modified the text as suggested as highlighted in the revised manuscript.

 L242-L249. What approach authors used to determine hyperparameters of the RF model for each class? Are they were selected manually, or authors used some ML tools? This part should also be clarified in this section.

Authors: Thanks for your question. They were selected based on the spectral characteristics of the LULC classes to be classified.  

Subsection 2.4. If authors used MapBiomas as a source of validation data – it should be described in this section.

Authors: Thanks for your suggestion. We included references of MapBiomas in the introduction and method sections in the revised manuscript. 

Methods for validation of the RF model should also be described in Section 2

Authors: Thanks for your suggestion. It was done as suggested in the revised manuscript. 

Figure 6, 7, 8 -  Image quality is too low, it will be a good idea to resave it in 300 dpi

Authors: Thank you for your suggestion. The Figures 6, 7, 8 were saved in 300 dpi as suggested.

I wish that my comment would be helpful in improving the quality of this research. Thank you.

Authors: Thanks very much for your comments that improved the revised manuscript.

Reviewer 3 Report (New Reviewer)

Dear Authors,

The authors present a new method to map Land Use and Land Cover (LULC) classes in the São Paulo State, Brazil, using Landsat-8 Operational Land Imager (OLI) data. The novelty of the proposed method consists of selecting the images based on the spectral and temporal characteristics of the LULC classes.

Based on the above, I have some comments

1. - The paper contain all main obligatory chapters (Introduction, Materials and Methods, Results, Discussion). This is Ok.

2. - The manuscript is well written, the structure, and the research questions the author poses they are well cared for.

3. - The introduction offers an overview of the state of the art, as well as the proposed methods.

However, the review part of Random Forest (RF) algorithm. (Line 73 - 77) is very poor and the citations are insufficient. The introduction would benefit from the following:

a. which other methods are used in literature for similar scope

b. advantages and disadvantage of the option

c. for which other applications these methods are used

4.- In methods (line 118 - 120), image processing includes radiometric and atmospheric corrections, what techniques were exactly performer? They could explain it better.

Author Response

Reviewer-3

Comments and Suggestions for Authors

The authors present a new method to map Land Use and Land Cover (LULC) classes in the São Paulo State, Brazil, using Landsat-8 Operational Land Imager (OLI) data. The novelty of the proposed method consists of selecting the images based on the spectral and temporal characteristics of the LULC classes.

Authors: Thanks very much for your positive comments. All comments and suggestions were considered and improved the revised manuscript.

Based on the above, I have some comments

  1. - The paper contain all main obligatory chapters (Introduction, Materials and Methods, Results, Discussion). This is Ok.

Authors: Thanks for your positive comment.

  1. - The manuscript is well written, the structure, and the research questions the author poses they are well cared for.

Authors: Thanks for your positive comment.

  1. - The introduction offers an overview of the state of the art, as well as the proposed methods.

However, the review part of Random Forest (RF) algorithm. (Line 73 - 77) is very poor and the citations are insufficient. The introduction would benefit from the following:

  1. which other methods are used in literature for similar scope
  2. advantages and disadvantage of the option
  3. for which other applications these methods are used

Authors: Thanks for comments and suggestions. We improved the text about RF in the revised manuscript.

4.- In methods (line 118 - 120), image processing includes radiometric and atmospheric corrections, what techniques were exactly performer? They could explain it better.

Authors: Thanks for your question and suggestion. These were explained better in the revised manuscript.

Round 2

Reviewer 2 Report (Previous Reviewer 1)

The manuscript have significantly improved from the last submussion, but figures resolution MUST be improved. Captions on Figures 2,4,5,7 and 8 are unreadble 

Author Response

Reviewer-2-REV

Comments and Suggestions for Authors

The manuscript have significantly improved from the last submission, but figures resolution MUST be improved. Captions on Figures 2,4,5,7 and 8 are unreadble 

Authors: Thanks very much for your positive comments and suggestions. Figures 2, 4, 5, 7 and 8 were improved as suggested.

This manuscript is a resubmission of an earlier submission. The following is a list of the peer review reports and author responses from that submission.

Round 1

Reviewer 1 Report

The effort of mapping LULC using Landsat data in Brazil using VFS method is interesting and will bring a significant contribution in this field.

Besides this, the manuscript needs serious improvements.

Introduction part looks small. It is a good idea to add more literature review about same cases in Brazil or São Paulo State. Methods also should be reviewed.

Table 1 should be better styled (size of the rows should be increased).

Table 2. Pasture class is missing in the first column

Map with samples locations should be added.

Figure 6. It is a good idea to present each class using Landsat data instead of Sentinel-2. Also it is a good idea to represent samples using true color composite (432).

Figure 8. It is a good idea to present each class as a separate column

References part is too small. Authors should make deeper literature analysis

I wish that my comment would be helpful in improving the quality of this research.

Thank you.

Author Response

Reviewer 1

Comments and Suggestions for Authors

The effort of mapping LULC using Landsat data in Brazil using VFS method is interesting and will bring a significant contribution in this field.

 Authors: Thanks very much for your positive comments.

Besides this, the manuscript needs serious improvements.

Introduction part looks small. It is a good idea to add more literature review about same cases in Brazil or São Paulo State. Methods also should be reviewed.

Authors: Thanks very much for your comment. The introduction part was improved including additional references in the revised manuscript.

Table 1 should be better styled (size of the rows should be increased).

Authors: Thanks very much for your suggestion. It was done in the revised manuscript.

Table 2. Pasture class is missing in the first column

Authors: Thanks very much for your observation. Pasture was indirectly classified as the last class to the classification mosaic. This was explained in the revised manuscript.

Map with samples locations should be added.

Authors: Thanks very much for your suggestion. We added the samples location in Figure 1 (Study area).

Figure 6. It is a good idea to present each class using Landsat data instead of Sentinel-2. Also it is a good idea to represent samples using true color composite (432).

Authors: Thanks very much for your suggestion. The Figure 6 was changed as suggested. We used Sentinel-2 only to collect the samples for validation as the spatial resolution is higher than Landsat-OLI. The samples used for validation were represented using true color composite, as suggested, in the revised manuscript.

Figure 8. It is a good idea to present each class as a separate column

Authors: Thanks very much for your suggestion. It was done in the revised manuscript.

References part is too small. Authors should make deeper literature analysis

Authors: Thanks very much for your suggestion. We have included more references to support our work.

I wish that my comment would be helpful in improving the quality of this research.

Authors: Thanks very much for your helpful comments that improved the manuscript.

Reviewer 2 Report

Please see my attached document

Author Response

Reviewer 2

This is an interesting paper that addresses an important problem of land use and land cover assessment using remote sensing. The research question is well defined and is a good fit to the journal scope.

Authors: Thanks very much for your positive comments.

The aims and objectives need to be more explicit to explain why this method is required = another data product is used for a comparison, why is this not good enough? What classification accuracy is needed? The novelty of the work is not discussed or justified. There are significant gaps in the description of the method, which makes it difficult to assess the merits of the work and it would not be easy for others to reproduce the method. The results are flawed in my opinion by making a comparison against another product, which will have its own inaccuracy, rather than comparing them both to the same reference data. This compromises the discussion and conclusions.

Authors: Thanks very much for your comments. The paper was improved following the comments and your suggestions as presented in the revised manuscript.

The quality of the manuscript is generally good, and it is easy to read and understand.

Authors: Thanks very much for your positive comments.

The presentation of the results needs some improvement at the detailed level as I have suggested below. The conclusions would be of wide interest to readers of the journal and would merit publication if the significant shortcomings are addressed.

Authors: Thanks very much for your helpful comments.

General concept comments

The manuscript is clear, well-structured, and is very easy to read.

Authors: Thanks very much for your positive comments.

I have provided some detailed comments and suggested improvements in the table, below. There are three major deficiencies in this manuscript that need to be addressed prior to publication:

  1. Gaps in the methods, in particular the creation of the image mosaics, the determination of spectral endmembers for LSMM, the training of the random forest classifier, and how conflict between the individual class classifications have been handled.

Authors: Thanks very much for your helpful comments and suggestions. We improved the manuscript with more details about the method as suggested.

  1. The comparison with the MapBiomass product is of little consequence as the reader has no insight from this which performs better. The two products should be compared against the same reference data and then the performance metrics compared.

Authors: Thanks very much for your helpful suggestion. We compared the two results using the same reference data and then the performance metrics were compared as suggested.

  1. The manuscript does not adequately explain what the novelty of the approach is.

Authors: Thanks very much for your helpful comments. We explained why the method we developed has novelty and originality through the text in the revised manuscript.

Despite these problems the work is of sufficient interest that the authors should be offered the opportunity to address them,

Authors: Thanks very much for your positive comments.

Specific Comments: Section      Line                      Comment

Abstract            17                           Authors talk about pre-processing the images to   highlight each LULC class. However, if this was true, what would be the purpose of the random forest classifier. I think authors mean that the images were pre-processed with other data to derive information expected to improve the classification of LULC.

Authors: Thanks very much for your comments. You are right, we pre-processed the images to highlight each LULC class. We used random forest classifier that is available in the GEE platform, but any other classifier can be used to generate similar results. You are right, we pre-processed with other data to derive information expected to improve the classification of LULC classes individually.

Introduction                                       28-32                     I think authors should explain what is meant by land use and land cover and the differences between them. For example, that agriculture is a land use, but sugarcane would be a land cover. The classes are mostly land use, but agriculture could include pasture, so you have a hybrid scheme, which is not a problem, but is worth explaining.

Authors: Thanks very much for your helpful suggestions. You are right about land use and land cover differences. We have explained this concept in the revised manuscript.

47                           You suggest that all the Atlantic Forest and Cerrado has been removed but then you go on to say how much. So you need to qualify by “mostly been removed” or similar.

Authors: Thanks for your suggestion. We agree with you and changed to “mostly been removed” as suggested.

54                           Phytophysionomic (I think drop the “s”)

Authors: Thanks for your suggestion. We agree with you and corrected it.

Materials and Methods                 90                           I think juts say RGB composite rather than include the band numbers. It will confuse readers not familiar with the band designations.

Authors: Thanks for your suggestion. We agree with you and changed to RGB without the band numbers in the revised manuscript.

95                           You need a reference here for the LSMM (you have it later)

Authors: Thanks for your observation. We included the reference for the LSMM as suggested.

97-103                  You need to provide more detail on how the mosaics were created. If it was a mosaic, how did you match the images radiometrically at the edges. Was it perhaps a composite. If not say why you did not use a compositing method. I suggest you add more detail to justify the time periods in each case, too. Why, for example, do you need an annual mosaic for water?

Authors: Thanks very much for your comments. We added details about the mosaic creation. Yes, it is a composite done using the GEE platform. We included more details about the time periods in each case as showed in the revised manuscript.

104                        There `is no scale on the maps

Authors: Thanks for your observation. We included the scale on the maps as suggested.

111                        Table 1 is badly formatted in my copy with overlapping lines. The layout needs to be corrected

Authors: Thanks for your observation. We improved the table format in the revised manuscript.

112-113                 The NTL data appears to come from a different source that you have not introduced in the text. I suggest you rename 2.2 as “remote sensing data”, Then add a short paragraph on the NTL data after current line 103 to explain what it is.

Authors: Thanks for your suggestion. We renamed 2.2 as Remote sensing data and added description of NTL at the end of table.

121-123                 Your method does not explain exactly how you obtained the spectral endmembers. How did you do the visual interpretation and was their sampling involved. Did the visual interpretation use a different source of data and does that matter?

Authors: Thanks for your comment. We have explained how to obtain the spectral endmember in the revised manuscript, i.e., the spectral endmembers (vegetation, soil, and shade/water) were selected by analyzing the spectral and temporal characteristics of these endmembers.

128                        Figure 3 does not include the training of the random forest classifier and how that was done.

Authors: Authors: Thanks for your observation. Figure 3 is corrected as suggested.

147                        Fig 5 should have scale and north arrow.

Authors: Thanks for your observation. We corrected Fig.4 and 5 as suggested.

150-151                 I would expect to see a description of the process for training the random forest classifier at this point. Was training data selected in a similar fashion to the validation data?

Authors: Thanks for your comment. We have added the description of the process for training the random forest classifier in the revised manuscript as suggested.

152-178                 You should explain in more detail the reasons for selecting the data for each class. Did you do any iteration on these to see what produced the best results, such as a sensitivity analysis? What is the significance of the percentiles – presumably showing the range of variability? Something structure like Table 3 would be informative.

Authors: Thanks for your comment. We have added the details about the selection of the data for each class and discussed it in the revised manuscript as suggested.

174                        If you are classifying separately for each class, I would expect there to be overlaps in the classification between the layers. For example, a pixel that is classified as both agriculture and pasture. How have you resolved these conflicts?

Authors: Thanks for your question. The overlaps problem is resolved by using the image mosaic composite where we started with the classes that are easier to classify with less errors. The composite algorithm avoids the overlapping classes. We included this explanation in the revised manuscript.

177                        Table 2 first line is missing the class in the first column.

Authors: Thanks for your observation. We corrected it.

Results                                                 213                        Fig 8. Given there is only one bar I would suggest a pie chart is more appropriate

Authors: Thanks for your suggestion. We agree with you and corrected it.

229-238                 I think you have the wrong approach for comparing your results with MapBiomas product. This is also a product derived from remote sensing and will have its own sources of inaccuracy. By comparing the two you have no idea which is best. I would like to see the MapBiomas (as prediction) compared to your reference data from interpretation of Sentinel-2 images. You will then get values of kappa and accuracy for MapBiomas against the same reference data. Comparing the performance of your approach with this will allow you to reach a conclusion about how much better or worse your method it to this product.

Authors: Thanks for your suggestion. We agree with you and followed your suggestion. We compared the Mapbiomas with our reference from interpretation of Sentinel-2 images and then compared the product obtained by both methods.

Discussion                                         270-271                 Your results only show that there is a good agreement with MapBiomas, you cannot say which is better or worse. If you change the analysis as suggested above, you will be able to comment on that.

Authors: Thanks for your suggestion. We agree with you and followed your suggestion.

Conclusion                                         278-280                 You are suggesting because MapBiomas has a more complex classification that it performs better than your method? But you could compare them against the same reference data as I suggest and be able to say which is better then explain why.

Authors: Thanks for your suggestion. We agree with you and followed your suggestion.

General                                                In the Intro, Discussion and Conclusions you do not explain why the method you have developed has novelty and originality

Authors: Thanks very much for your helpful comments. We explained why the method we developed has novelty and originality through the text in the revised manuscript.

Reviewer 3 Report

Mapping Land Use Land Cover Classes in São Paulo State, Southeast of Brazil, using Landsat-8 OLI Multispectral Data and Visual Feature Selection Method

Title: It is quite broad. This title must be reducing their expectations to what they do, and better explain the “new” components of your paper. At this point, this is more a technical report than a scientific paper.

Abstract: First need to explain why it is necessary to develop a new method. The objective is not clear, must include the “method”. Then, the “method” must be clearly presented. Everything is diffuse here. Results are not clear, and it is more technical and descriptive, than a scientific point of view. Nothing new were presented here. You need to clarify this in the abstract. Las sentence were not derived from the results presented in the abstract. Here you must present a conclusion derived from your own outputs. You need to present clear data, and not just said “the method allows to minimize...”, according to what?

Introduction: It is very poor. This not explain the gap in the science that the paper wants to fill. Not explain what is new, and not explain the problem that want to solve. The objective presentation is very poor (lines 61-69). This is a summary of methods. We need here a clear objective, and some questions to be solved for this research.  This is not a technical report, this is a scientific journal. The objectives are not clearly linked to the title.

Methods: They are OK depending on what you need to do. The methods are not linked to the objectives nor to the title. We need to know WHY you do all of this? This is not a clear justification. This is more like a technical report where you put everything that you made, and not a clear proposal of methods based on the needs of ONE objective that you propose first.

Results: The results are totally descriptive. There are no comparisons with other methods, and not answer any clear scientific question.

Discussion: This section is quite poor, and are not linked to the results, and most of the discussion are not linked to the outputs presented in the paper.

Conclusions: Here you need to present the NEW knowledge derived from your results for the Science. You present a summary of your technical obtained data.

In summary, paper must be better structured and presented following a SCIENTIFIC purpose. I think, data is quite valuable, but the draft must be re-structured to improve the quality. At this time, this kind of paper are not attractive for the readers of this Journal.

Author Response

Reviewer-3

Title: It is quite broad. This title must be reducing their expectations to what they do, and better explain the “new” components of your paper. At this point, this is more a technical report than a scientific paper.

Authors: Thanks very much for your comments. We changed the title as suggested. The paper was improved following your comments and suggestions.

Abstract: First need to explain why it is necessary to develop a new method. The objective is not clear, must include the “method”. Then, the “method” must be clearly presented. Everything is diffuse here. Results are not clear, and it is more technical and descriptive, than a scientific point of view. Nothing new were presented here. You need to clarify this in the abstract. Las sentence were not derived from the results presented in the abstract. Here you must present a conclusion derived from your own outputs. You need to present clear data, and not just said “the method allows to minimize...”, according to what?

Authors: Thanks very much for your comments. The paper was improved following your comments and suggestions as presented in the revised manuscript.

“The novelty of the proposed method consists of selecting the images based on the spectral and temporal characteristics of the LULC classes”.

Method: “First, we defined six classes to be mapped in the year 2020 as forest, forest plantation, water bodies, urban areas, agriculture and pasture. Second, we visually analyzed the spectral characteristics of these targets over the year to select the image dates that better represent each class in the images. Then we pre-processed these images to highlight each LULC class. Finally, the classification was performed using the Random Forest classifier algorithm, available in the Google Earth Engine (GEE) platform, for each class individually and then the final map was composed by all selected LULC classes”.

“Therefore, the method allows to minimize classification errors by facilitating post-classification edition of the individual mapped classes”

Introduction: It is very poor. This not explain the gap in the science that the paper wants to fill. Not explain what is new, and not explain the problem that want to solve. The objective presentation is very poor (lines 61-69). This is a summary of methods. We need here a clear objective, and some questions to be solved for this research.  This is not a technical report, this is a scientific journal. The objectives are not clearly linked to the title.

Authors: Thanks very much for your comments that improved the manuscript. We changed the title as suggested and now the objectives are directly linked to the title.

Title: “Mapping Land Use Land Cover Classes in São Paulo State, Southeast of Brazil, using Landsat-8 OLI Multispectral Data and Derived Spectral Indices and Fraction Images”

Objectives: “this work aims to map the LULC in the state of São Paulo using the Landsat-8 Operational Land Imager (OLI) images acquired mainly during the year 2020, and the de-rived spectral indices and fraction images, based on the classification of individual LULC classes: forest, pasture, agriculture, urban areas, and water”.

Methods: They are OK depending on what you need to do. The methods are not linked to the objectives nor to the title. We need to know WHY you do all of this? This is not a clear justification. This is more like a technical report where you put everything that you made, and not a clear proposal of methods based on the needs of ONE objective that you propose first.

Authors: Thanks very much for your comments. We changed the title as suggested and now the method is directly linked to the title and the objectives.

Results: The results are totally descriptive. There are no comparisons with other methods, and not answer any clear scientific question.

Authors: Thanks very much for your comments. The results were improved following the comments and your suggestions.

Discussion: This section is quite poor, and are not linked to the results, and most of the discussion are not linked to the outputs presented in the paper.

Authors: Thanks very much for your comments. The discussion was improved following your comments and suggestions.

Conclusions: Here you need to present the NEW knowledge derived from your results for the Science. You present a summary of your technical obtained data.

 Authors: Thanks very much for your comments. The conclusions were improved following your comments and suggestions.

In summary, paper must be better structured and presented following a SCIENTIFIC purpose. I think, data is quite valuable, but the draft must be re-structured to improve the quality. At this time, this kind of paper are not attractive for the readers of this Journal.

Authors: Thanks very much for your comments and suggestions that improved the revised manuscript.

Round 2

Reviewer 2 Report

I wish to to thank the authors for the time taken to address my concerns with the original submission. These have mostly been resolved. The performance of the MapBiomass product that you now present is very good. However this raises the question about why another product or method is required. I don't think you have expressed this very clearly in your conclusions.  The significance of the results is not very clear as a result.

Author Response

I wish to thank the authors for the time taken to address my concerns with the original submission. These have mostly been resolved. The performance of the MapBiomass product that you now present is very good. However this raises the question about why another product or method is required. I don't think you have expressed this very clearly in your conclusions.  The significance of the results is not very clear as a result.

Authors – We thank you very much for your comments and suggestions that improved the revised manuscript. We acknowledged you and other reviewers (anonymously) in the revised manuscript. We showed that our results and MapBiomas results are very good and comparable. We also mentioned that our method is simpler than MapBiomas method and produced the similar results in the conclusions.
